# Biosynthetic Nanobubble-Mediated CRISPR/Cas9 Gene Editing of Cdh2 Inhibits Breast Cancer Metastasis

**DOI:** 10.3390/pharmaceutics14071382

**Published:** 2022-06-30

**Authors:** Ruru Gao, Qiong Luo, Yang Li, Liming Song, Junnan (Stephen) Cai, Ying Xiong, Fei Yan, Jianhua Liu

**Affiliations:** 1Department of Medical Ultrasound, The Second Affiliated Hospital, School of Medicine, South China University of Technology, Guangzhou 510180, China; 13607043754@163.com (R.G.); 13044206218@163.com (Q.L.); xyzjy09154013@163.com (Y.X.); 2Department of Gastrointestinal Surgery, Shenzhen People’s Hospital (The Second Clinical Medical College, Jinan University), Shenzhen 518020, China; szrmyyyy@126.com; 3Department of Gastrointestinal Surgery, Shenzhen People’s Hospital (The First Affiliated Hospital, Southern University of Science and Technology), Shenzhen 518020, China; 4Department of Orthopedics, Zhujiang Hospital, Southern Medical University, Guangzhou 510282, China; songlimingsmu@163.com; 5SXUltrasonic (Shenzhen) Ltd., Shenzhen 518000, China; stephenchoi@sxultrasonic.com; 6Center for Cell and Gene Circuit Design, CAS Key Laboratory of Quantitative Engineering Biology, Shenzhen Institute of Synthetic Biology, Shenzhen Institutes of Advanced Technology, Chinese Academy of Sciences, Shenzhen 518055, China

**Keywords:** ultrasound, CRISPR/Cas9, gene editing, epithelial–mesenchymal transition, N-cadherin

## Abstract

The epithelial-mesenchymal transition (EMT), a process in which epithelial cells undergo a series of biochemical changes to acquire a mesenchymal phenotype, has been linked to tumor metastasis. Here, we present a novel strategy for knocking out the EMT-related Cdh2 gene, which encodes N-cadherin through CRISPR/Cas9-mediated gene editing by an ultrasound combined with biosynthetic nanobubbles (Gas Vesicles, GVs). Polyethyleneimine were employed as a gene delivery vector to deliver sgRNA into 4T1 cells that stably express the Cas9 protein, resulting in the stable Cdh2 gene- knockout cell lines. The Western blotting assay confirmed the absence of an N-cadherin protein in these Cdh2 gene-knockout 4T1 cell lines. Significantly reduced tumor cell migration was observed in the Cdh2 gene-knockout 4T1 cells in comparison with the wild-type cells. Our study demonstrated that an ultrasound combined with GVs could effectively mediate CRISPR/Cas9 gene editing of a Cdh2 gene to inhibit tumor invasion and metastasis.

## 1. Introduction

Breast cancer is one of the most prevalent malignant tumors in women, and its incidence has increased faster than other cancers [1,2]. Specially, triple-negative breast cancer (TNBC) is the most aggressive subtype of breast cancer, with a 10–20% incidence [3]. The invasion and metastasis of TNBC are greatly promoted by epithelial–mesenchymal transition (EMT), a process in which epithelial cells undergo a series of biochemical changes to acquire a mesenchymal phenotype [4]. The key molecular processes during EMT are downregulation of E-cadherin (E-cad, Cdh1) and upregulation of N-cadherin (N-cad, Cdh2) [4,5]. The over-expression of N-cadherin has been strongly correlated with poor patient prognosis [6]. In addition, the lack of N-cadherin expression in tumor cells can significantly inhibit breast tumor cell invasion and migration.

Gene-editing technology mediated by the clustered regularly interspaced short palindromic repeats (CRISPR)-associated protein 9 (CRISPR/Cas9) has gained increasing attention in recent years and has emerged as one of the most promising methods for the treatment of various diseases [7,8]. To accomplish their gene-editing function, the CRISPR/Cas9 systems must be delivered into the cells. To date, there are many delivery vectors which are used to deliver CRISPR/Cas9 including viral and non-viral vectors [9]. The viral vectors have some obvious drawbacks, such as a complicated procedure and high immunogenicity. By contrast, the non-viral vectors, such as polyethyleneimine (PEI), cationic liposomes, and various nanoparticles, show some unique advantages, including improved safety, greater flexibility, and more facile manufacturing. Electrostatic interaction allows the cationic substance to form a complex with anion DNA, which may then be taken up by cells through endocytosis [10,11]. Although non-viral vectors outperform viral vectors in terms of biological safety, their transfection efficiency is significantly lower. Furthermore, both viral and non-viral vectors are unable to deliver CRISPR/Cas9 in a time-and space-controllable manner. Therefore, it is necessary to develop a novel gene delivery approach to address these issues.

Ultrasound-targeted microbubble destruction (UTMD) is a recently developed technology successfully used for gene or drug delivery and has recently attracted wide attention in biomedical applications. UTMD possesses many advantages over conventional gene delivery strategies, such as non-invasiveness, low cost, targeted delivery, and repeatability [12,13,14,15]. Acoustic cavitation effects from bubbles excited by ultrasound irradiation were mostly implicated in the process. Microbubbles would oscillate and rupture upon being irradiated by ultrasonic energy, producing mechanical forces such as shock waves, microstreaming, and micro-jetting to surrounding tissues or blood vessels. These mechanical forces further cause reversible holes on the surface of the targeted cells and increase cell membrane permeability, promoting drug/gene penetration into the target cells [15]. However, the particle size of conventional microbubbles is on a micron-scale. They have difficulty with directly contacting tumor cells to produce cavitation effects due to the presence of a vascular wall, resulting in low gene delivery efficiency. Luckily, nanobubbles with nanoscale particle size have recently been developed. The biosynthetic nanobubbles (gas vesicles, GVs) from *Halobacterium NRC-1* or *Anabaena flos-aquae* have proteinaceous shells which are similar to the cell membrane, endowing them with good biocompatibility and safety [16,17]. In vitro and in vivo investigations have demonstrated that GVs can produce stable ultrasonic contrast signals and may be used as gene carriers to deliver genes [18,19]. In this study, we utilized the biosynthetic GVs to deliver the plasmid containing sgRNA targeting the exon3 of the Cdh2 gene to Cas9-stably expressed 4T1 breast cancer cells via ultrasound-mediated bubble cavitation, followed by functional verification of the inhibition in tumor invasion and metastasis (Figure 1).

## 2. Materials and Methods

### 2.1. Cell Culture

4T1 cells were purchased from ATCC (Manassas, VA, USA) and modified according to the needs of the experiment to stably express Cas9 protein and enhanced green fluorescent protein (EGFP), which was termed CRISPR Cas9 4T1-Cas9-hyg stable cell line (SL581; GeneCopoeia, Inc., Rockville, MD, USA). The gene-modified 4T1 cells were cultured in the RPMI 1640 medium (Corning, New York, NY, USA) supplemented with 10% fetal bovine serum (Gibco, 10099141C, New York, NY, USA) and 1% penicillin–streptomycin (Corning, 30-002-CI) incubated at 37 °C under 5% CO_2_.

### 2.2. Bacterium Culture and Extraction of GVs

*Halobacterium* (*Halo*) was cultivated, and the GVs were extracted according to a previous report [18]. Briefly, *Halo* was cultured in the medium for 2 weeks (centrifuged at 100–120 rpm at 42 °C). The bacterial culture was transferred into a separatory funnel and maintained for 1 week to allow the bacteria to float to the top of the medium. The hypotonic shock method was used to lyse these bacteria. Subsequently, the bacterial lysate was centrifuged at 300× *g* for 3–5 times at 4 °C (each for 4 h) to isolate the GVs. The upper milky-white layer of GVs was collected and then resuspended in PBS for subsequent experimentation (4 °C, avoid vibration). The concentration of GVs was determined by measuring the absorbance at 500 (OD500) by using a microplate reader (BioTek Synergy 4, Winooski, VT, USA).

### 2.3. Preparation and Characterization of GVs-PEI-DNA (GPD)

Polyethyleneimine (1 g/L, 25 KDa, branched PEI, Sigma-Aldrich, Saint Louis, MO, USA) was diluted to the working concentration (1 μg/μL, pH 7.0). First, the isolated GVs (OD500 = 1.0) in PBS were centrifuged at 300× *g* for 0.5 h at 4 °C, and the lower layer solution was gently removed with a 1-mL syringe. Then, 6 μg PEI in PBS solution were added into the GVs and mixed gently at the volume ratio of 10:1, and then left standing for 40 min at 37 °C. Next, the samples were centrifuged at 200× *g* for 45 min at room temperature. The lower layer solution which contained free PEI was removed with a 1-mL syringe. Finally, the floated cationic GVs (GV–PEI complex) were obtained. After that, 4 μg plasmid DNA containing the sgRNA cassette was added to the GVs–PEI complex, followed by incubation for 30 min at 37 °C to obtain the GPD transfection complex. Then, the plasmid DNA containing the sgRNA cassette was added to the GVs–PEI complex at the ratio of 1: 1.5 (mass ration), which was incubated for 30 min at 37 °C to obtain the GPD transfection complex. The particle size and zeta charge of GVs or GPD were analyzed using the Zetasizer NANO ZS System (Malvern, UK) at room temperature.

### 2.4. In Vitro Ultrasound-Mediated Gene Transfection

Ultrasound-mediated gene transfection was performed using an ultrasound transfection instrument (Sonovitro^®^ MAN Ultrasound Transfection System, MAN-IG, Shenzhen, China). Briefly, 1.5 × 10^5^ cells were seeded into 24-well plates (Corning, 3524) and cultured overnight. When the density of each well reached 70–80%, the culture plate was washed once with PBS and then replaced with a fresh complete medium. The GPD transfection complex was added at the concentration of 3 μg plasmid/well. The ultrasound irradiation was performed with the following parameters: 1 MHz transmit frequency, 1 W/cm^2^ power, 20% duty cycle, and 1 min duration. Next, the cells were incubated in a complete culture medium for 4 h. Then, the cells were washed thrice with PBS and incubated for another 24–96 h. The fluorescence was observed through inverted fluorescence microscopy (Olympus IX7, Tokyo, Japan), and the cells were collected for subsequent flow cytometry and sorting.

### 2.5. sgRNA Sequences and Vector Construct

One sgRNA targeting an EGFP reporter gene sequence and two sgRNAs targeting the Cdh2 gene were designed and synthesized by GeneCopoeia Technologies, with the following sequence: GGGCGAGGAGCTGTTCACCG/sg-EGFP; GATTTCAAGGTGGACGAGGA/sg-Cdh2-1; CCGGGAGCCAACCCTGACTG/sg-Cdh2-2. U6 promoter and sgRNA scaffold were amplified from the PX458 vector and inserted into the PciI site of pCMV-C-mCherry. Both EGFP and Cdh2 sgRNAs were cloned into the modified pCMV-C-mCherry under the U6 promoter via homologous recombination to obtain the pU6-sgRNA (EGFP)-mCherry and pU6-sgRNA (Cdh2)-mCherry plasmids.

### 2.6. Determination of Gene Transfection and Editing Efficiencies

To determine the ultrasound-mediated gene transfection efficiency, the following groups were included: (1) control group; (2) GPD group; (3) GPD + US group. For convenient observation, pU6-sgRNA (EGFP)-mCherry was used. After 48–72 h of gene transfection, the cells were collected and resuspended in 400 μL PBS. Then, the transfection efficiency was determined by flow cytometry (BD FACSAria™ III, BD, Franklin Lakes, NJ, USA). The data are expressed as the ratio of the red mCherry fluorescent cells in all green fluorescent cells. The successfully transfected cells (emitting red fluorescence) were separated by flow sorting and further cultured until the EGFP fluorescence disappeared due to the occurrence of gene editing of EGFP. After 21 days, the gene-editing efficiency was determined through flow cytometry by counting the ratio of 4T1 cells that did not emit EGFP fluorescence to the mCherry-expressed 4T1 cells.

### 2.7. CCK-8 Assay

Cytotoxicity was analyzed through the CCK-8 assay (DOJINDO, Shanghai, China) according to the manufacturer’s protocols. Approximately 10^4^ cells contained in a 100 μL medium were seeded into a 96-well plate. After 24 h of cultivation, the transfection complex of the control, GVs–PEI, and GPD and lipofectamine 2000 were added into the 96-well microplates. The cells transfected with GPD were collected with or without ultrasound irradiation. Then, 10 μL of the CCK-8 reagent was added to each well at 6 h, 12, or 24 h after the transfection. The absorbance was analyzed at 450 nm by using a microplate reader (BioTek Synergy 4).

### 2.8. Western Blotting Assay

The expression of N-cad protein in the 4T1 cells and Cdh2-knockout (Cdh2-KO) cells was determined through Western blotting. The membrane proteins were extracted from 4T1 cells and Cdh2-KO cells by using the RIPA lysis buffer (Beyotime, P0013, Shanghai, China), and the protein concentration was determined using a BCA protein assay kit (Beyotime, P0012). The samples were separated on a 10–12% SDS-PAGE gel in the Mini-PROTEAN Tetra System (Bio-Rad, Hercules, CA, USA). Then, the proteins were transferred onto polyvinylidene fluoride (PVDF) membranes by using the Mini-PROTEAN Tetra System. After that, the samples were blocked for 15 min in a blocking solution (Beyotime, P0252). Next, the membrane was incubated with anti-N-cadherin antibody (Abcam, ab76011, Cambridge, UK) and anti-GAPDH antibody (Beyotime, AF1186) at 4 °C overnight, and incubated with horseradish peroxidase-conjugated secondary antibodies for 1 h. After rinsing thrice, enhanced chemiluminescence (ECL, Millipore, WBKlS0100, Burlington, MA, USA) detection reagents were added to the membrane. Protein signals were read using a chemiluminescent imaging system (Tanon 5200, Shanghai, China). ImageJ software was used to analyze the results.

### 2.9. In Vitro Wound Healing Assay and Cell Migration Experiments

The wild-type 4T1 cells (1.5 × 10^5^) and two Cdh2-KO cell lines (named No.3 and No.59) were seeded into 24-well plates. When the cell density of each well reached 80–90%, the wound healing assay was performed. Briefly, a vertical line was drawn from the upper edge of each well. After that, PBS was added to wash out the unadhered cells. The cells were then incubated under standard culture conditions. The extent of wound healing at different time points was observed by inverted microscopy. The wound healing area was calculated by ImageJ software.

The Transwell experiments were also performed to determine the cell migration ability. In brief, the wild-type, No.3, and No.59 4T1 cells (1.5 × 10^4^) were seeded into the upper chamber (Corning, 3422) with 200 μL serum-free medium and incubated overnight. After 12, 24, or 36 h, the cells that had migrated to the lower surface of the filter were stained with 1% crystal violet (Sigma-Aldrich, V5265) and then photographed. The stained cells were counted by ImageJ software.

### 2.10. Animals and Tumor Model

All experimental animals were obtained from the Beijing Vital River Laboratory Animal Technology Co., Ltd. (Beijing, China). Briefly, BALB/c mice (age: 4–6 weeks, weight: 14–16 g, female gender) were used to establish tumor models. In addition, 1 × 10^6^ 4T1 cells or Cdh2-knockout (Cdh2-KO) cells were suspended in 100-μL serum-free RPMI1640 medium and injected into the fourth pair of sub-mammary fat pads to generate an in situ breast tumor model. The tumor volume and the mouse body weight were measured every 2 days until the mice were sacrificed after 30 days. The tumor volume (V) was calculated according to the following formula: V = (length × width^2^)/2, where the length represents the longest diameter of the tumor and the width represents the shortest diameter. The relative tumor volume (RTV) was calculated as follows:RTV = V/V1
where V1 represents the first measurement of the tumor volume. The lungs of the mice were harvested to count the number of lung tumor nodules.

In addition, the metastatic tumor model was also established through tail intravenous injection of 1 × 10^5^ tumor cells. The weight of the mice was recorded every 2 days for 14 days. After 2 weeks, the lungs of the mice were harvested to count the number of lung nodules and stained with H&E after sectioning. All factors that may induce clinically relevant abnormalities among the animals were carefully avoided during the study.

### 2.11. Statistical Analysis

Data are represented as the mean ± SEM. One-way analysis of variance with Fisher’s least significant difference test and two-way ANOVA was applied for multiple comparisons using Prism 8 (GraphPad Software). A *p*-value of < 0.05 was considered to indicate statistical significance.

## 3. Results

### 3.1. In Vitro Ultrasound-Mediated Gene Editing of EGFP

The Cas9- and EGFP-expressed 4T1 breast cancer cells were utilized to test the feasibility of ultrasound-mediated gene editing in vitro. To make GVs be able to bind with plasmids, we modified these negatively charged GVs with PEI. The resulting GV–PEI complexes had 17.90 ± 2.39 mV zeta potential, significantly higher than the plain GVs (−21.63 ± 0.97 mV). Notably, a larger particle size was also observed in these GV–PEI complexes, with 386.90 ± 9.15 nm mean particle size versus 259.90 ± 3.09 nm plain GVs (Appendix A). GVs binding with pU6-sgRNA (EGFP)-mCherry plasmid (GVs–PEI–DNA, GPD) were obtained (Appendix A), and then incubated with Cas9- and EGFP-stably expressed 4T1 cells, followed by irradiation with or without ultrasound. The cells treated with only pU6-sgRNA (EGFP)-mCherry plasmid were used as a control. Figure 2A illustrates that the GPD + US group had more red fluorescent cells than the GPD group, indicating that GVs combined with ultrasound irradiation might promote gene transfection. Quantitative analysis by flow cytometry showed that the transfection efficiency of the GPD + US group was 10.31% ± 1.35%, significantly higher than that of the GPD group (4.92% ± 0.26%) and the control group (0.05% ± 0.01%) (Figure 2B,C). Figure 2D shows that the EGFP editing efficiency of the GPD + US group was significantly higher than that of the GPD and control groups.

### 3.2. In Vitro Ultrasound-Mediated Gene Editing of Cdh2

Next, we used this technology to knock out Cdh2 in the Cas9- and EGFP-stably expressed 4T1 breast cancer cells. Two expression cassettes containing different sgRNAs targeting Cdh2 gene sequences were inserted into the pU6-sgRNA-CMV-mCherry vector to increase the gene editing efficacy. The resulting pU6-sgRNA (Cdh2)-mCherry plasmid was mixed with PEI-modified GVs before being introduced to the Cas9- and EGFP-stably expressed 4T1 cells. Figure 3A shows that the GPD + US group had significantly more mCherry-expressed cells than the groups without ultrasound irradiation or untreated groups. Flow cytometry analysis showed that the transfection efficiency of the GPD + US group was 12.27% ± 0.54%, which was significantly higher than that of the GPD group (6.28% ± 0.34%) and the control group (0.05% ± 0.01%) (Figure 3B,C). In addition, the transfection efficiency of US group was significantly higher than that of the lipo2000 group (Appendix A). These results revealed that ultrasound in conjunction with GVs can increase the delivery efficiency of pU6-sgRNA (Cdh2)-mCherry plasmids and allow the mCherry gene to express in the 4T1 cells. We acquired 63 Cdh2-edited cell lines by DNA sequencing after isolating the mCherry-expressed cells through flow sorting and single-cell clonal culture. Overall, 13 cell lines were successfully edited in both sgRNA1 and sgRNA2 targeting sites, whereas 22 cell lines in the sgRNA1 site and 29 cell lines in the sgRNA2 site. The result of Sanger sequencing was presented in Appendix A, showing the successful gene editing. The Western blotting analysis confirmed the loss of N-cadherin expression in the No.3 and No.59 dual-site-edited cells (Figure 3D).

### 3.3. Invasion Behavior Assay of Cdh2 KO Cells

The wound healing assay was performed using No.3 and No.59 cell lines to evaluate the invasion behavior of Cdh2 KO cells. As a control, the wild-type 4T1 cells (WT) were employed. Figure 4A shows that the WT cells could heal the wound after 12 h and completely heal after 24 h. By contrast, both the No.3 and No.59 groups showed a significantly slower wound healing capability than the WT group. Quantitative analysis revealed that the area recovery rate of the WT group was 7.73% ± 1.07% and 78.72% ± 1.39% at 12 h and 24 h, respectively. In contrast, the area recovery rate of the No.3 group was only 0.95% ± 0.33% and 9.13% ± 0.58% at 12 h and 24 h, respectively. Similarly, the area recovery rate of the No.59 group was 1.49% ± 0.10% and 6.35% ± 0.66% at 12 h and 24 h, respectively (Figure 4B). The transwell experiment was used to assess the cell migration ability of these Cdh2 KO cells. Figure 4C shows that the No.3 and No.59 groups had significantly fewer cells in the lower chamber at 12, 24, and 36 h than the WT group. In the No.3 and No.59 groups, the quantitative analysis revealed that only 30 and 40 cells per view filed were migrated to the bottom chamber. However, there were 300 cells per view in the bottom chamber for the WT group (Figure 4D). Thus, our results indicated that ultrasound-mediated GVs-mediated gene editing of Cdh2 may effectively suppress the migration of 4T1 cells.

### 3.4. In Vivo Tumor Metastasis of Cdh2-Edited Cells

The Cdh2-KO cells were then tested in vivo to see if they might inhibit tumor metastasis. The in situ breast cancer model was established by injecting the WT cells, No.3, or No.59 cells into the fat pad of mammary glands on the right side of the mouse (Figure 5A). The size of the tumor and the body weight of mice were monitored every two days after tumor transplantation. The mice were sacrificed on the 30th day, and the lung tissue was harvested to count the metastatic pulmonary nodules. The results showed there were significantly more metastatic pulmonary nodules in the WT group than in the No.3 and No.59 groups (Figure 5B). Quantitative analysis of lung sections showed that the average number of pulmonary nodules in the WT group was 12.80 ± 1.59, much higher than those in the No.3 group (1.60 ± 0.68) and No.59 group (3.00 ± 0.84) (Figure 5C). Statistical analysis showed that there were significant differences between the control group and Cdh2-KO groups. In addition, we found that the WT group exhibited a faster rate of tumor growth than the No.3 and No.59 groups (Figure 5D). The lung metastasis capability of Cdh2-KO cells was evaluated by intravenously injecting these tumor cells into the mice. The mice were sacrificed after 14 days, and the lung tissue was harvested to count the metastatic pulmonary nodules (Figure 6A). Our results showed that the WT group had a number of metastatic pulmonary nodules, whereas the No.3 and No.59 groups just had a few nodules (Figure 6B). The quantitative results showed that the WT group had approximately 76.20 ± 6.92 tumor nodules, while the No.3 and No.59 groups had only 6.00 ± 2.03 and 2.40 ± 0.93 tumor nodules, respectively (Figure 6C). Notably, as compared with the No.3 and No.59 groups, the WT group had significantly reduced body weight (Figure 6D). These findings indicated that knockout of Cdh2 by gene editing may inhibit the metastasis of 4T1 cells.

## 4. Discussion

In this study, we successfully developed a new method for achieving ultrasound-mediated gene editing by using PEI-modified GVs as a DNA-binding gene delivery vector. After ultrasonic stimulation, the plasmid containing the sgRNA system may be delivered into the cells that can perform Cas9-mediated gene editing. Cas9-mediated gene editing is widely used for the treatment of diseases including cancer, AIDS, leukemia, sickle cell disease (SCD), β-Thalassemia, Duchenne muscular dystrophy, and other diseases due to the precise deletion or repair of specific genes at the genome level [20,21,22,23,24]. However, most delivery systems have flaws, such as immunogenicity, instability, low delivery efficiency, and lack of targeting, limiting their application in disease treatment [25,26]. In this study, we demonstrated the viability of gene editing by using cationic GVs and ultrasound to deliver CRISPR/Cas9 components. Ultrasound-mediated gene editing has many clear advantages. First, the physical electrostatic interaction allows these cationic GVs to bind with DNA encoding CRISPR/Cas9 components, preventing enzyme digestion. Second, the cavitation effects generated by these GVs excited by ultrasound may perforate the nearby cell membrane, favoring the delivery of CRISPR/Cas9 components into the cells. Third, it may be used to deliver CRISPR/Cas9 components locally at deep-seated diseased sites since ultrasound has a high tissue penetration capability.

Ultrasound has recently been proved to be a non-invasive, repeatable, and targeted method for improving gene transfection efficiency [15,27]. However, ultrasound has only been used in a few studies for CRISPR/Cas9-mediated gene editing. The size of the bubble is one of the key reasons. Usually, bubbles as cavitation nuclei can be divided into micron-scale microbubbles and nanoscale nanobubbles. The large size of microbubbles limits them to exude out of the blood vessels into tissue, preventing them from directly cavitating against tumor cells and delivering DNA into the tumor cells. The nanoscale GVs were utilized as a DNA delivery vector in our research. Nanobubbles have a relatively smaller particle size than the traditional microbubbles, endowing them with higher stability, larger surface area, and tissue permeation capabilities. GVs can load therapeutic DNA and more effectively deliver them into tumor cells through ultrasound cavitation effects by altering their surface with cationic PEI [28,29]. Recently, Cai et al. used the CRISPR/Cas9-C-erbB-2 plasmid to microbubble-transfect human endometrial cancer (HEC)-1A cells. Their results proved that CRISPR/Cas9-C-erbB-2 delivery through microbubble reduced C-erbB-2 protein expression in HEC-1A cells [30]. However, its transfection efficiency was significantly lower than that of GVs-mediated gene transfection.

In our study, a cationic polymer PEI was used to modify biosynthetic nanobubbles (GVs) as a gene carrier to deliver sgRNA into 4T1 cells that stably express Cas9 protein. To the best of our knowledge, this is the first time that biosynthetic nanobubbles (GVs) and PEI have been combined as a novel non-viral delivery system for delivering sgRNA via the cavitation effect of ultrasound for gene-editing applications. According to our findings, the EMT-related gene Cdh2 was successfully deleted by GVs-mediated sgRNA delivery. Significant reduced invasion and metastasis could be observed in the Cdh2-deleted cells in vitro and after these cells were introduced in vivo. The high expression of N-cad protein in 4T1 cells may play an important role in tumor invasion and metastasis by regulating epithelial-to-mesenchymal transition. As a result, our study provided a non-invasive, non-viral gene approach for delivering CRISPR/Cas9 components for gene-editing applications. Notably, numerous important genes, including novel biomarkers such as circSETD3 and NPTX1, are still engaged in tumor invasion and metastasis [31,32]. Considering the advantages of ultrasound, such as non-invasiveness, targeting, and repeatability, the future in situ gene editing in the in vivo condition might be performed using GVs-meditated local delivery technology.

## 5. Conclusions

In this study, we introduced a GVs-mediated gene transfection approach to deliver the CRISPR/Cas9 system for gene-editing applications. The EGFP and Cdh2 genes were successfully knocked out in the 4T1 cells. Importantly, Cdh2-KO cells showed significantly reduced invasion and metastasis behavior, and slower tumor growth. Given the advantages of ultrasound in terms of non-invasiveness and the focusing capability, our study paved the path for future in situ gene-editing studies in vivo.

## Figures and Tables

**Figure 1 pharmaceutics-14-01382-f001:**
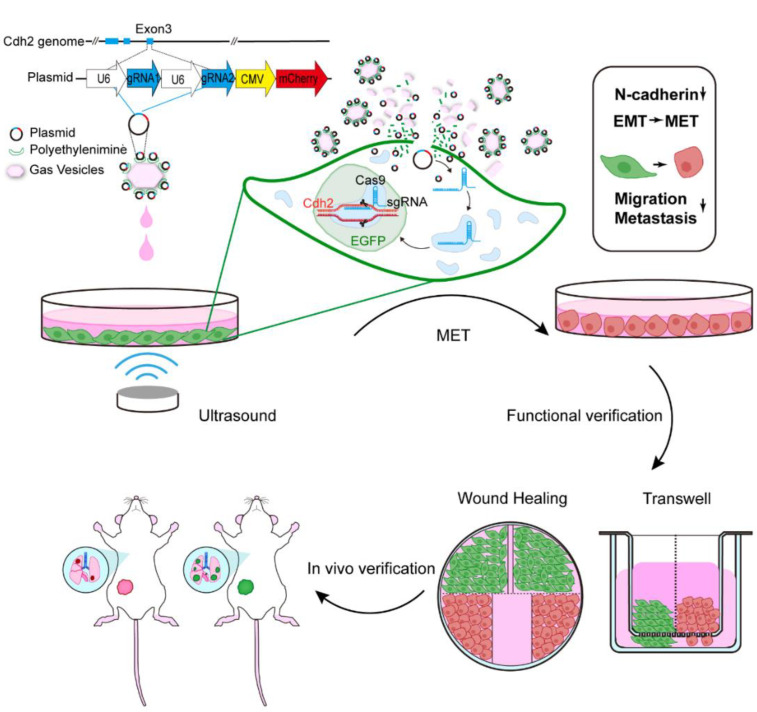
Schematic diagram of ultrasound-mediated CRISPR/Cas9 gene editing of Cdh2 to inhibit tumor invasion and metastasis.

**Figure 2 pharmaceutics-14-01382-f002:**
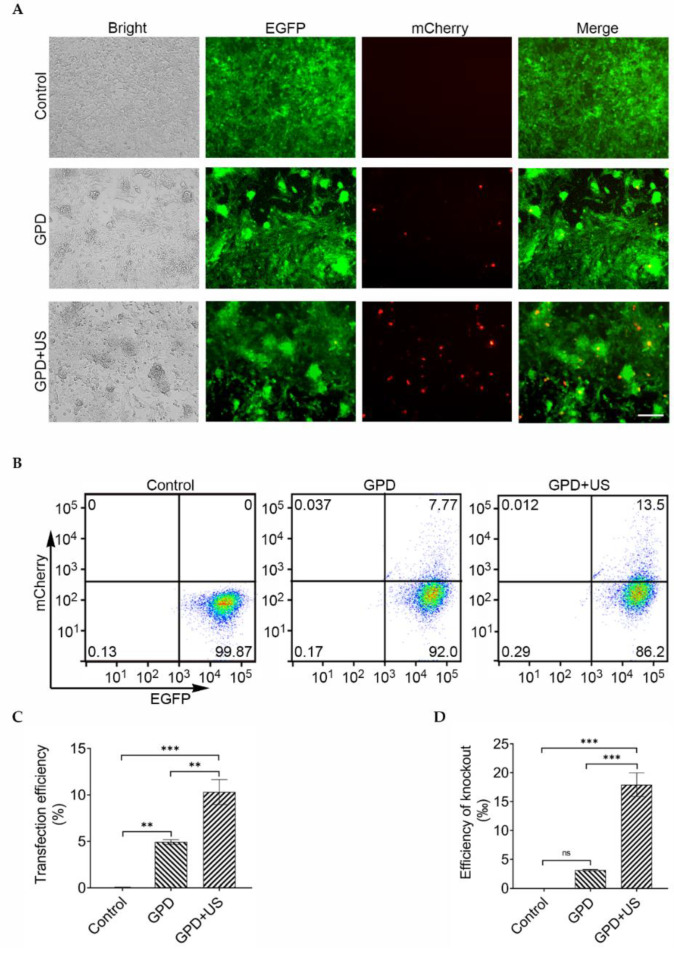
(**A**) Fluorescence images of Cas9- and EGFP-stably expressed 4T1 cells transfected with only pU6-sgRNA (EGFP)-mCherry (control), GPD, or GPD + US (GPD: the abbreviation of GVs-PEI-DNA). Scale bar = 200 μm; (**B**) determination of the transfection efficiency by flow cytometry through counting of the red-emitting cells; (**C**) quantitative analysis of the mCherry-expressed cells from (**B**) (*n* = 3); (**D**) determination of the EGFP knockout efficiency (*n* = 3). ns denotes *p* > 0.05, ** *p* < 0.01, *** *p* < 0.001.

**Figure 3 pharmaceutics-14-01382-f003:**
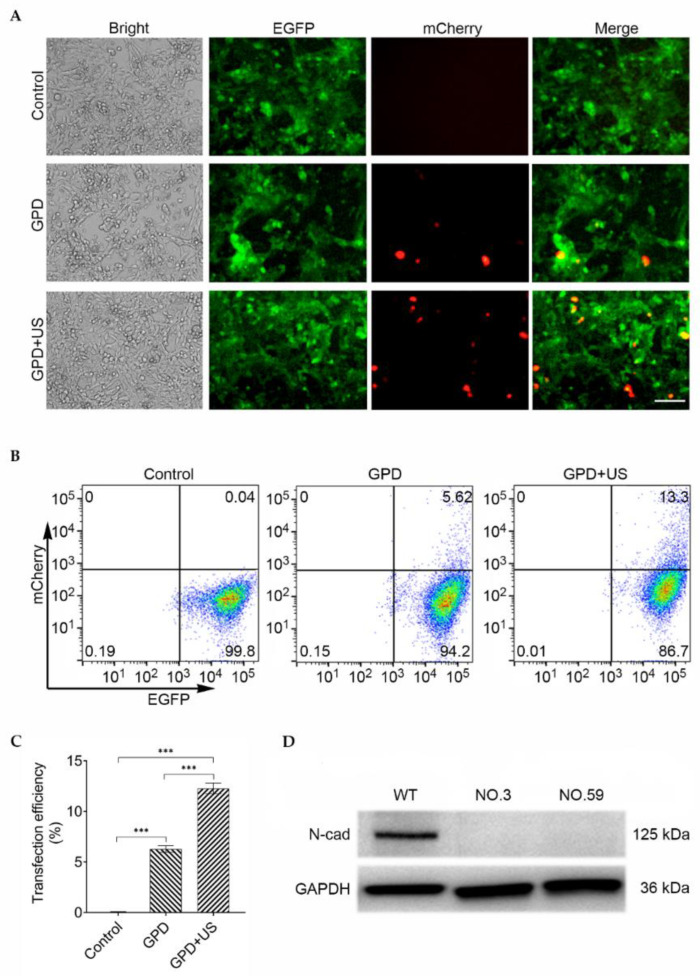
(**A**) Fluorescence images of Cas9- and EGFP-stably expressed 4T1 cells transfected with only pU6-sgRNA (Cdh2)-mCherry (control), GPD, or GPD + US (GPD: the abbreviation of GVs–PEI–DNA). Scale bar = 100 μm; (**B**) determination of the transfection efficiency by flow cytometry through the counting of the red-emitting cells in EGFP-expressed cells; (**C**) quantitative analysis of the mCherry-expressed cells from (**B**) (*n* = 3) *** *p* < 0.001. (**D**) Western blotting assay of the expression of N-cadherin in the wild-type (WT) and two Cdh2-KO cell lines. GAPDH was used as an internal marker.

**Figure 4 pharmaceutics-14-01382-f004:**
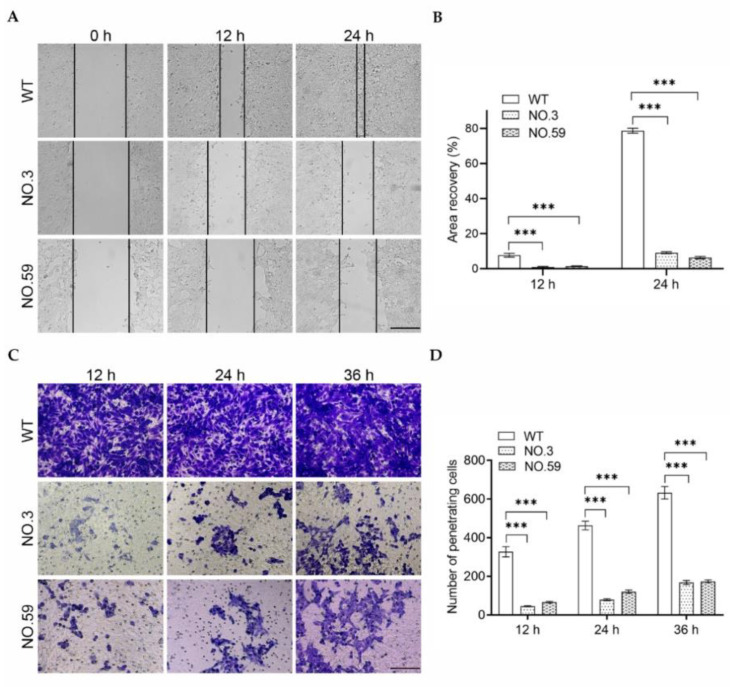
(**A**) Wound healing assay of the wild-type and Cdh2-KO cell lines at 0, 12, or 24 h. Scale bar = 150 μm; (**B**) quantitative analysis of the recovery area of the cells from (**A**). (*n* = 3); (**C**) transwell cell migration assay of the wild-type and Cdh2-KO cell lines at 12, 24, or 36 h. Scale bar = 150 μm; (**D**) quantitative analysis of the cells migrated to the bottom chamber from (**C**) (*n* = 3). *** *p* < 0.001.

**Figure 5 pharmaceutics-14-01382-f005:**
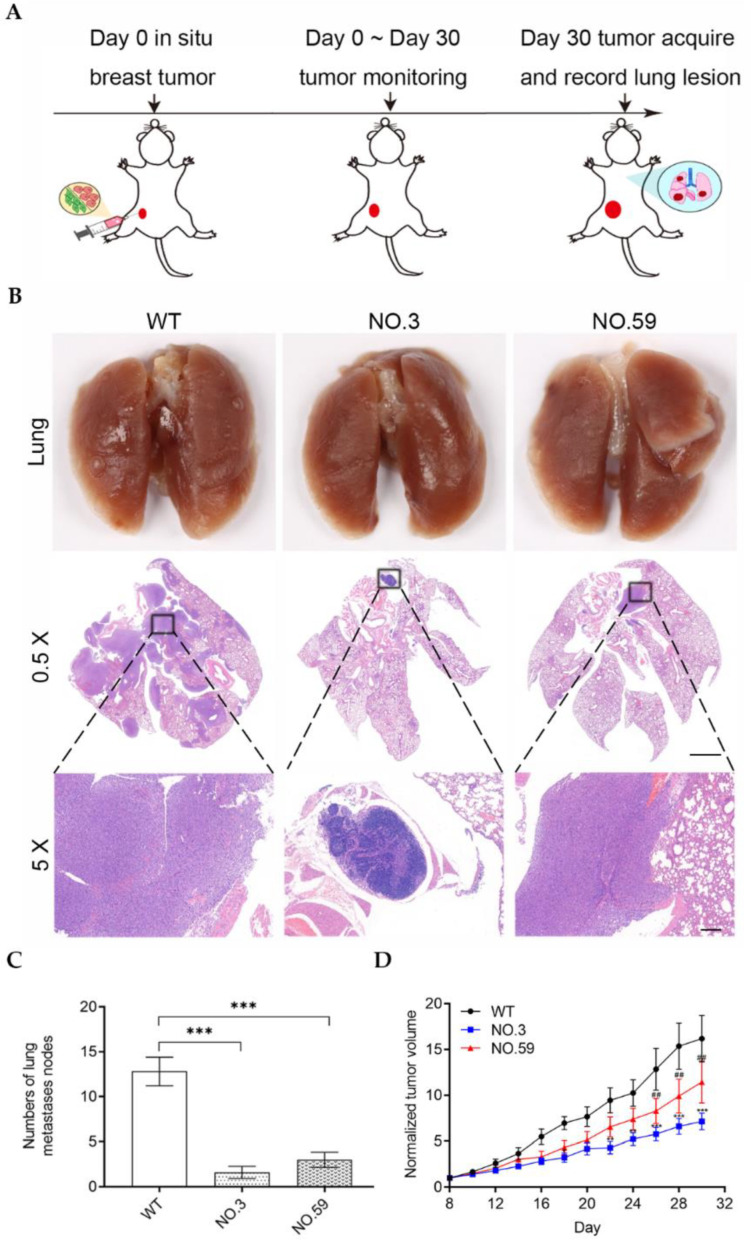
(**A**) Schematic illustration of the mouse experimental procedure for assessing the pulmonary metastasis capabilities of the Cdh2-KO cells in the orthotopic tumor model; (**B**) photographs of the lungs and the H&E staining of the lung sections of mice injected with the wild-type or Cdh2-KO cell lines. 0.5× magnification, Scale bar: 2 mm. 5× magnification, Scale bar: 200 μm. (**C**) The lung nodules were counted under an anatomical microscope. (*n* = 5); (**D**) orthotopic tumor growth curves of different groups of tumor-bearing mice. (*n* = 5). ‘*’: WT vs. No.3, ‘#’: WT vs. No.59. ** or ## denotes *p* < 0.01, *** *p* < 0.001.

**Figure 6 pharmaceutics-14-01382-f006:**
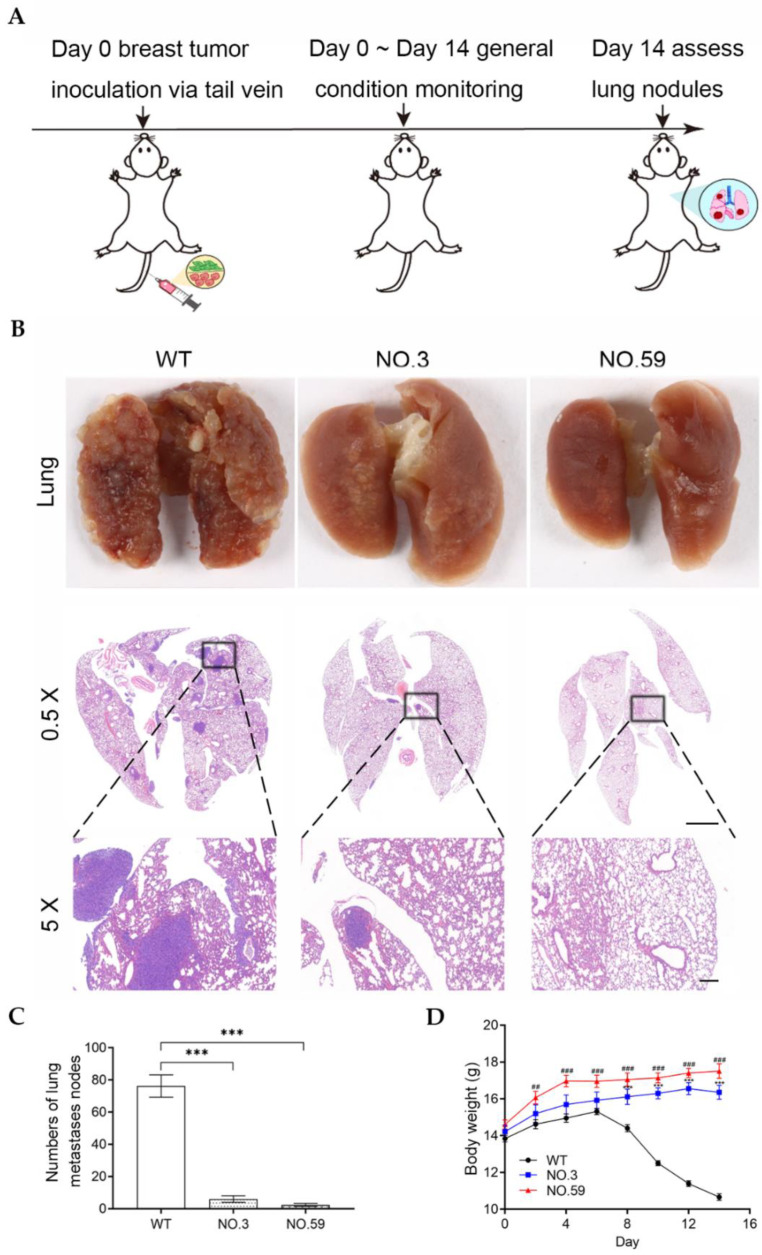
(**A**) Schematic illustration of the mouse experimental procedure for assessing the pulmonary metastasis capabilities via tail intravenous injection of the Cdh2-KO cells; (**B**) photographs of the lungs and H&E staining of the lung sections of mice injected with the wild-type or Cdh2-KO cell lines; magnification: 0.5×, Scale bar: 2 mm. Magnification: 0.5×, Scale bar: 200 μm. (**C**) The lung nodules were counted under an anatomical microscope. (*n* = 5); (**D**) bodyweight change curves of the three groups of mice. (*n* = 5). ‘*’: WT vs. No.3, ‘#’: WT vs. No.59. ## denotes *p* < 0.01, *** or ### *p* < 0.001.

## Data Availability

Not applicable.

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
