# Peer review of "Biosynthetic Nanobubble-Mediated CRISPR/Cas9 Gene Editing of Cdh2 Inhibits Breast Cancer Metastasis"

_pharmaceutics, 2022, doi:10.3390/pharmaceutics14071382_

Round 1

Reviewer 1 Report

Title: Biosynthetic nanobubble-mediated CRISPR/Cas9 gene editing of Cdh2 inhibits breast cancer metastasis.

In this paper, the authors study breast cancer in women. The authors use ultrasound to combat metastases. The authors conclude that the study demonstrated that ultrasound combined with gas vesicles (GV) could effectively mediate CRISPR / Cas9 gene editing of the Cdh2 gene to inhibit tumor invasion and metastasis.

The very complex figure 1, 5 and 6 are not sufficiently explained.

  • This paper presents some news, but for early clinical diagnosis and intervention, the authors should mention the new biomarker . In this regard, below I report an interesting article that should be studied, incorporate their meaning and report them briefly in the discussion and in the list of references.

New biomarker for lung cancer - focus on circSETD3.

Tao F, Gu C, Li N, Ying Y, Cao LF, Xiao QF, Ni D, Zhuang YB, Zhang Q.J Biol Regul Homeost Agents. 2021 Mar-Apr;35(2):583-591.

In addition, recently, in an article it was reported that the downregulation of NPTX1 induces cell cycle progression through Wnt / β-catenin signaling in breast cancer. In this regard, below I report an interesting article that should be studied, incorporate their meaning and report them briefly in the discussion and in the list of references.

Downregulation of NPTX1 induces cell cycle progression through Wnt/β-catenin signaling in breast cancer.

Ye RY, Kuang XY, Shao N, Wang SM, Lin Y.J Biol Regul Homeost Agents. 2021 May-Jun;35(3):1177-1183.

I believe these suggestions are important for improving this paper. Without these corrections the paper cannot be published. So I recommend minor revision.

Reviewer 2 Report

Major issues 

There are several issues that should be clarified:

In Figure 3 B and C, the brightfield images are of poor quality - compared to Control group, it is difficult to detect individual cells. Photomicrographs from the ‘green channel’ should also be provided.

Figure 3 D, for the WB photomicrograph, the kDA should be provided.

Figure 5A - What is ‘tumor supervise’?

Page 2, line 82 starts: ‘Our previous studies have shown…’ and is followed by references 18 and 19. However, none of the authors in either manuscript overlap. What do the authors mean by ‘our previous studies’?

Line 112-118 - more details of the method/reference should be provided. E.g. how was the PEI ‘removed’?

Line 364-367 - this sentence gives an impression that the gene-editing was performed as a therapeutic strategy in vivo, which is not the case, and, thus, it should be re-phrased.

Minor issues:

The manuscript should undergo thorough English language proofreading.

Examples of linguistic and stylistic imperfections: 

Line 21, 46 - ‘evidences’. The word ‘evidence’ has no plural.

Line 22 vs Line 41 - transformation vs transition should be unified.

Line 33 - mediate

Line 123 - remove ‘for’

Line 145 - ‘emit’

Line 172 - PBS was ‘added’

Line 174 - how?

Line 195 - missing verb

Line 211 - ‘from this picture’ - such phrases should be omitted

Line 214 - ‘quantitative analysis by flow analysis’

Line 272 - ‘obviously’ - such words should be avoided

Line 339/340 - attribute

Line 364 - ‘and so on’ - such phrases should be avoided.

Reviewer 3 Report

The authors have developed an in vitro gene-editing method using nanobubbles. I think this study looks like Pharmacology rather than Pharmaceutics. As a Pharmaceutics study, it is necessary to develop in vivo transfection methods. In addition, the stability of formulation in terms of transfection activity, particle size, and so on should be tested. And, comparisons with positive control about transfection efficiency and cell toxicity are required to indicate the superior capacity of the vector.

Author Response

请参阅附件。

Reviewer 4 Report

The manuscript entitled "Biosynthetic nanobubble-mediated CRISPR/Cas9 gene-editing of Cdh2 inhibits breast cancer metastasis" by Gao et al. utilized GVs to deliver the CRISPR/Cas9 system for editing Cdh2 gene in 4T1 breast tumor cells through ultrasound-mediated bubble cavitation for inhibiting their invasion and metastasis. Though this is a relevant article for pharmaceutics readers, several issues prevent publication in its current format. My primary concerns are:

  1. Figure 2: The overall transfection efficiency is very poor. The authors should use commercial transfection agents such as lipofectamine or Fugene as a positive control. For GPD and GPD+US groups, cells are not in monolayer, and a portion of mCherry expression is not associated with the cells (Figure 2A). Please provide a higher resolution Figure. Figure 2D: Please mention the number of replicates for each group.
  2. Figure 3: Please provide fluorescent images since the cells were stably expressed. Also, add a panel of DAPI stained cells for each group for better comparison. Why is there a large difference in transfection efficiency in Figure 2 and Figure 3?
  3. Please provide the number of replicates for each quantitative data.
  4. Figure 5: It is unclear at which time points there is a significant difference in tumor growth among various groups. Please clearly indicate the exact time point by placing a significant symbol on a respective day.

Round 2

Reviewer 2 Report

Comments #2:

Original comment 1: In Figure 3 B and C, the brightfield images are of poor quality - compared to Control group, it is difficult to detect individual cells. Photomicrographs from the ‘green channel’ should also be provided.

Response: We apologized if our original Figure 3 did not show the cells clearly. According to your suggestion, we have provided a group of images containing the green channel to replace the previous set of images.

Response 1#2: Now, the brightfield images are even smaller, and it is even more difficult to recognize an individual cell.

Original comment 4: Page 2, line 82 starts: ‘Our previous studies have shown…’ and is followed by references 18 and 19. However, none of the authors in either manuscript overlap. What do the authors mean by ‘our previous studies’?

Response 4: Thank you for pointing this out. We have made a revision. (Page 2, line 84).

Response 4#2: I suggest that the authors avoid such ‘mistakes’ in the future, and comply with the scientific ethical standards.

Original comment 5: Line 112-118 - more details of the method/reference should be provided. E.g. how was the PEI ‘removed’?

Response 5: We removed the free PEI through centrifugal floatation method at 200 g for 45min at room temperature. The GVs-PEI would be floated and solution containing the free PEI was removed by a syringe. We added more details of the method in the revised version. (Page 4, line 121~129; Figure S1)

Response 5#2: It is still unclear, how the procedure was executed. Taking into account the standard terms used in centrifugation, such as supernatant and pellet - if the GVs-PEI were ‘floated’ and the PEI were in a solution, that is supernatant, how were they removed by the syringe? Was there no pellet? What is the difference between centrifugal floatation and standard centrifugation? 

Original comment 7:  ‘The manuscript should undergo thorough English language proofreading.

Examples of linguistic and stylistic imperfections:...’

Response 7: ‘Thank you for your careful review. We have conducted a comprehensive and detailed examination of the article in English. According to your suggestions, we have made the following revisions:...’ 

Response 7#2:  As stated above, these were only examples of the issues. The manuscript should be further assessed by a professional English proofing service.

Reviewer 3 Report

Although the authors added Fig. S1 and S2, there were no effective descriptions for them in the manuscript. In vitro transfection efficiency with low cytotoxicity seemed to be good. However, the particle size of the vector was too large to achieve in vivo gene editing, especially against tumor tissues.

The transfection efficiency was far from 100%. Is such a vector promising for in vivo application in the case of cancer therapy?

About the targeting issue, the transfection efficiency of the vector without US irradiation would be too high to prevent gene editing against normal cells. Therefore, that without US should be decreased.

There was no evidence of gene editing, i.e. region specificity on the genome. If the locations other than the target region are edited, off-target effects such as increased malignancy may be a problem.

siRNA-based knockdown technologies have been developed. Is it necessary to knock out the gene in case of tumor cells?

Reviewer 4 Report

The authors have adequately addressed my concerns, and it could be accepted for publication.

Author Response

Thank you for your review. Best wishes!

Round 3

Reviewer 2 Report

Thank you for the corrections; the manuscript is now improved and acceptable for publishing in Pharmaceutics.